

# Effect of carbon nanoparticles on the growth and photosynthetic property of *Ficus tikoua* Bur. plant

Nian Chen[1,*], Xiaojian Tian[1,*], Mingli Yang[1], Jiajun Xu[1], Tinghong Tan[2] and Jiyue Wang[1]

[1] Key Laboratory of Surveillance and Management of Invasive Alien Species in Guizhou Education Department, Platform for Exploitation and Utilization of Characteristic Plant Resources, College of Biological and Environmental Engineering, Guiyang University, Guiyang, Guizhou, China

[2] School of Agriculture and Forestry Engineering and Planning, Guizhou Key Laboratory of Biodiversity Conservation and Utilization in the Fanjing Mountain Region, Tongren University, Tongren, China

[*] These authors contributed equally to this work.

## ABSTRACT

The application of nanomaterials in different plants exerts varying effects, both positive and negative. This study aimed to investigate the influence of carbon nanoparticles (CNPs) on the growth and development of *Ficus tikoua* Bur. plant. The morphological characteristics, photosynthetic parameters, and chlorophyll content of F. tikoua Bur. plants were evaluated under four different concentrations of CNPs. Results indicated a decreasing trend in several agronomic traits, such as leaf area, branching number, and green leaf number and most photosynthetic parameters with increasing CNPs concentration. Total chlorophyll and chlorophyll b contents were also significantly reduced in CNPs-exposed plants compared to the control. Notably, variations in plant tolerance to CNPs were observed based on morphological and physiological parameters. A critical concentration of 50 g/kg was identified as potentially inducing plant toxicity, warranting further investigation into the effects of lower CNPs concentrations to determine optimal application levels.

## INTRODUCTION

Carbon nanoparticles (CNPs) have received significant research interest in recent years due to their various applications. When introduced to plant systems, they can penetrate roots and translocate to shoots. Their ability to enter plant systems depends on factors such as size, concentration, solubility, plant species, and growth medium properties. CNPs have a profound impact on plant growth and development, enhancing biomass productivity, germination rates, and physiological processes (*Li et al., 2024*). By modulating gene expression, CNPs also alter plant molecular mechanisms. Furthermore, they play a crucial role in mitigating stress, ultimately contributing to the overall health and resilience of plants (*Aqeel et al., 2022*). Studies have shown carbon nanomaterials can boost seed germination and plant growth rates. *Jiwanti et al. (2022)* reported that carbon nanotube (CNT) penetration triggers changes in plant metabolic functions, leading to increased

Corresponding authors
Tinghong Tan, tthsqzx@126.com
Jiyue Wang, acute2803764@126.com

biomass, fruit production, and grain yield in some cases. Research also indicated CNT can stimulate plant photosynthetic efficiency, gene and protein expression, and production of metabolites such as compounds with medical value (*Chen et al., 2014*; *Giraldo et al., 2014*). Multi-walled carbon nanotubes (MWCNT) and graphene nanomaterials have demonstrated promise under stress conditions, for example by mitigating negative impacts of salinity and drought in various plant species. MWCNT have also been used to encapsulate fungicides as antifungal agents (*Fan et al., 2018*). An algal growth test was developed to determine effects of pristine and oxidized CNT on the green algae *Chlorella vulgaris* and *Pseudokirchneriella subcapitata* (*Schwab et al., 2011*). The results showed that the reduced algal growth might be caused mainly by indirect effects, *i.e.,* by reduced availability of light and different growth conditions caused by the locally elevated algal concentration inside of CNT agglomerates. Carbon-based nanomaterials further affect soil microorganisms, altering rhizosphere-root interactions (*Chen et al., 2015*). These effects can be either beneficial or detrimental to soil microbial communities. Recent studies have shed light on the impact of nanoparticles on crop growth under stressed environments. *Chen et al. (2023)* found that graphene nanoparticles enhance alfalfa growth through multiple metabolic pathways under salinity stress. A meta-analysis by *Li et al. (2023)* further revealed that carbon-based nanomaterials (CNMs) significantly boost shoot weight and antioxidant metabolites in cereal crops, but inhibit phytohormone secretion. The effects of CNMs vary depending on type, exposure duration, and crop variety, with short-term exposure negatively affecting roots and photosynthesis, and prolonged exposure having a promoting effect.

In addition, some studies reported that carbon nanomaterials may have some adverse influence on plant. *Intrchom et al. (2018)* demonstrated the toxicity of CNT metal hybrids on freshwater algae. CNT-Ag at a concentration of 5.0 mg/L showed 90% growth inhibition and also showed a significant effect on photosynthetic yield with a 21% drop compared to the control. *Shen et al. (2010)* have demonstrated that single-walled carbon nanotubes (SWCNTs) induce oxidative stress in *Arabidopsis* protoplasts and leaves, triggering programmed cell death. Despite the potential benefits of nanomaterials, their effects on plants require thorough evaluation. In particular, the influence of CNP on *Ficus tikoua* Bur. remains unknown.This study investigated the effects of CNPs on the morphological feature and photosynthetic parameters of *Ficus tikoua* Bur. plant, a fruit with medicinal and culinary uses prevalent in southwest China. This work aimed to elucidate the interactions between nanomaterials and plants, with the ultimate goal of informing the practical application of CNPs in agricultural production, thereby providing valuable insights for sustainable and efficient farming practices

## MATERIALS AND METHODS

### Materials

The CNPs used in this study were spherical with an average diameter of 40 nm. They were purchased from Beijing Dk nano S&T Ltd, China, and had a specific surface area of 200 $m^2$/g, bulk density of 0.09 $g/cm^3$, and true density of 0.43 $g/cm^3$. *Ficus tikoua* Bur.

was planted at the experimental field of Guiyang College in 2018. Stems and leaves were collected from this tree for vegetative propagation *via* cuttings, allowing reproduction of this variety for further study.

## Methods

In April, 2022, Cuttings of *Ficus tikoua* Bur. were planted in plastic buckets (28.0 cm depth × 26.0 cm diameter) containing 2 kg of nutrient-enriched soil. Different dosages of CNPs (50, 100, and 200 mg/kg) were added to the soil before planting. The plant was not exposed to CNPs as the control. Each treatment was replicated eight times for a total of 32 pots in a randomized block design experiment conducted in the test field at Guiyang University.

Growth characteristics of *Ficus tikoua* Bur. were assessed after one year of planting, inluding plant height (PH), stem diameter (SD), leaf area (LA), branching number (BN), leaf number (LN), green leaf number (GL). Photosynthetic parameters including photosynthetic rate (A), transpiration rate (E), intercellular $CO_2$ concentration (Ci), total conductivity to water vapor (gtw), and water use efficiency (WUE), Leaf chamber temperature (T), leaf-to-air vapor pressure deficit (VPD) and chamber relative humidity (RHcham) were analyzed using a GFS-3000 portable photosynthesis measuring device (LI-COR, USA) according to method described by *Sun et al. (2021)*. Each sample underwent four repeated measurements in the work.

All data were presented as the mean ± standard deviation of triplicate samples and analyzed using one-way analysis of variance followed by Duncan's test in SPSS 16.0. Treatment effects were deemed significant at $P < 0.05$. The subordinate function value method was employed to comprehensively assess plant resistance to CNPs (*Chen & Song, 2005*).

## RESULTS

### The effect of CNPs on growth of *F. tikoua* Bur. plant

Under different concentrations of CNPs, the changes of morphological characteristics of *F. tikoua* Bur. plants were presented in Fig. 1. Leaf area (LA), branching number (BN), and green leaf number (GL) exhibited a decreasing trend with increasing CNPs concentration. Notably, this trend was not observed at 50 g/kg CNPs concentration, as the difference from the control group was not found to be statistically significant ($P < 0.05$). Conversely, plant height (PH), stem diameter (SD), and leaf number (LN) remained unaffected by CNPs exposure.

### The effect of CNPs on photosynthesis of *F. tikoua* Bur. plant

Photosynthetic properties of *F. tikoua* Bur. plants exposed to CNPs were evaluated in Fig. 2. Except for WUE and RHcham, all photosynthetic parameters (E, A, Ci, VPD, T, and gtw) exhibited a decreasing trend with increasing CNPs concentration. However, E, A, WUE, gtw, T and RHcham of plants exposed to 50 g/kg CNPs showed no significant difference compared to the control group. In particular, RHcham in plants treated with 200 g/kg CNPs was significantly higher than that in the control group.

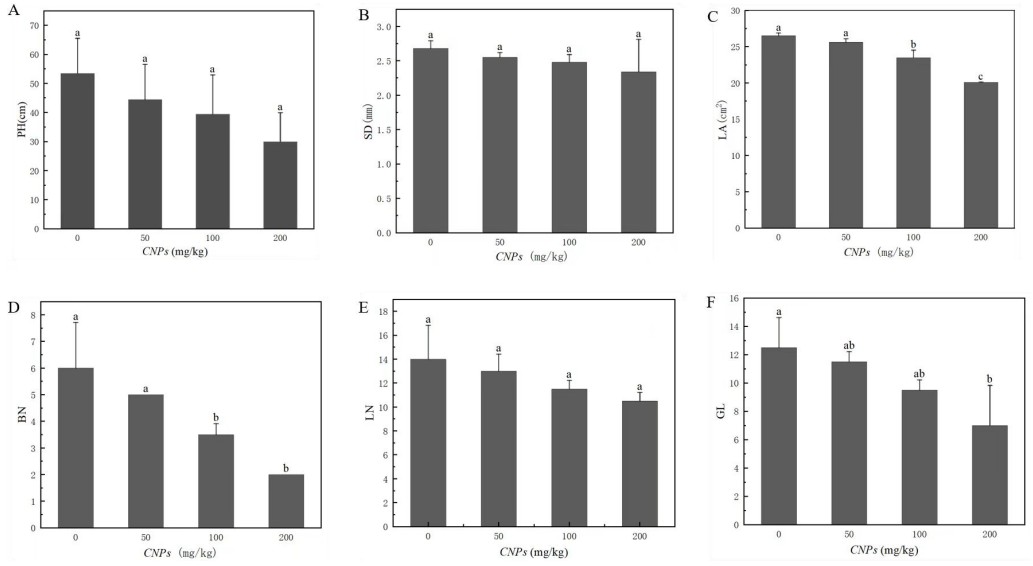

**Figure 1** **(A–F) Agronomic characteristics of *F. tikoua* Bur. plan exposed to CNPs.** Illustration, PH, plant height; SD, stem diameter; LA, leaf area; BN, branching number; LN, leaf number; GL, green leaf number. Different lowercase letters showed the significant differences among CNPs treatments at $p <$ 0.05. The same below unless particularly stated.

## The effect of CNPs on the leaf chlorophyll contents of *F. tikoua* Bur. plant

The results presented in Table 1 showed the impact of CNPs on leaf chlorophyll contents in *F. tikoua* plants. CNPs exposure had no remarkable influence on chlorophyll a (Chl a) levels. But, total chlorophyll (Chl t) and chlorophyll b (Chl b) contents were significantly lower in CNPs-exposed plants *versus* the control. Meanwhile, the Chl a/b ratio was significantly improved in CNPs-treated plants relative to the control. Intriguingly, there were no statistically significant differences observed between plants exposed to different CNP treatment levels. The data demonstrated that CNPs exposure can influence chlorophyll contents in *F. tikoua* leaves, with effects on specific chlorophyll parameters that differ between treatment and control samples.

## Correlation analysis

Correlation analysis revealed significant positive correlations among most agronomic traits in *F. tikoua* Bur. plants exposed to CNPs, except for SD with BN, LA and LN (Fig. 3A). Additionally, Fig. 3B reported that E exhibited a significant positive correlation with A and VDP, while A showed significant correlations with all indicators except for gtw and RHcham. VPD was found to have a significant positive correlation with E, A, Ci, and WUE. Moreover, WUE displayed significant positive correlations with A, Ci, VPD, and T. However, RHcham exhibited a significant negative correlation with E, A, and Ci. Clear correlations were observed in Fig. 3C between Chl t and Chl a, as well as Chl b, while the Chl a/b ratio had a negative correlation with Chl t and Chl b.

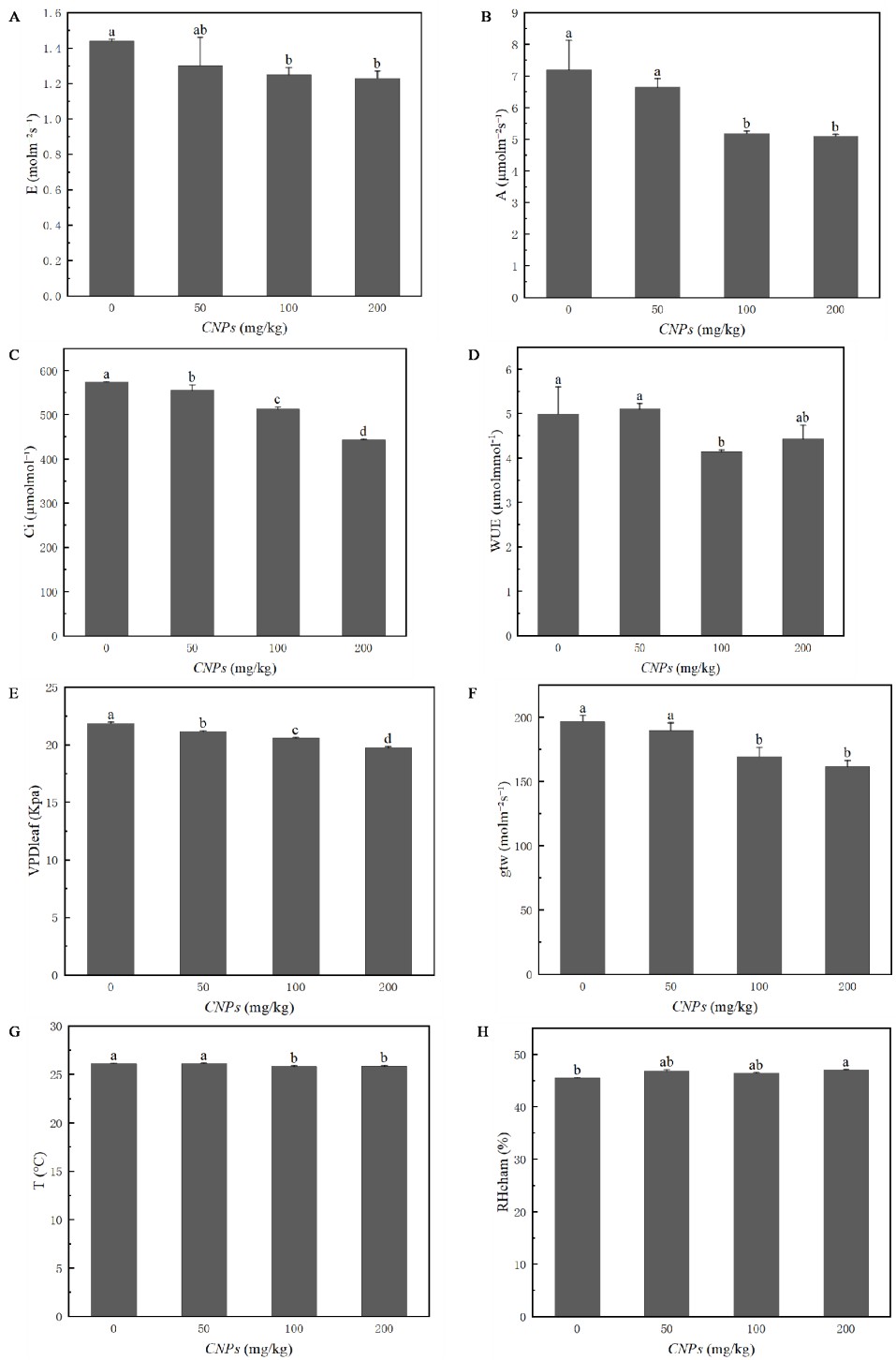

**Figure 2** (A–H) Photosynthesis parameters of *F. tikoua* Bur. plant exposed to CNPs.

**Table 1  The chlorophyll abundance of *F. tikoua* Bur. plant exposed to CNPs.**

| Index | CNPs (mg/kg) | | | |
|---|---|---|---|---|
| | 0 | 50 | 100 | 200 |
| Chl a (mg/g F.W) | 27.75 ± 0.32a | 27.56 ± 0.35a | 27.67 ± 0.03a | 27.25 ± 0.17a |
| Chl b (mg/g F.W) | 21.64 ± 0.54a | 16.8 ± 0.76b | 16.46 ± 0.14b | 16.13 ± 0.48b |
| Chl t (mg/g F.W) | 49.39 ± 0.84a | 44.36 ± 1.1b | 44.13 ± 0.17b | 43.39 ± 0.65b |
| Chl a/b ratio | 1.28 ± 0.02b | 1.64 ± 0.05a | 1.68 ± 0.01a | 1.69 ± 0.04a |

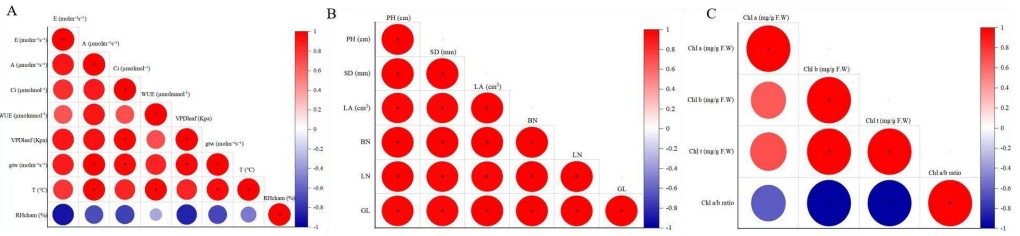

**Figure 3  The correlation coefficient of agronomical traits (A), photosynthetic parameters(B), chlorophyll indexes.**

**Table 2  The function values of agronomic trait in *F. tikoua* Bur.plant exposed to CNPs.**

| CNPs (mg/kg) | Function values of indicators | | | | | | Sum | Order |
|---|---|---|---|---|---|---|---|---|
| | PH | Leaf | SD | GL | BN | LN | | |
| 0 | 1 | 1 | 1 | 1 | 1 | 1 | 1 | 1 |
| 50 | 0.62 | 0.92 | 0.62 | 0.82 | 0.75 | 0.71 | 0.74 | 2 |
| 100 | 0.4 | 0.72 | 0.42 | 0.45 | 0.38 | 0.29 | 0.44 | 3 |
| 200 | 0 | 0 | 0 | 0 | 0 | 0 | 0 | 4 |

## Resistance evaluation

In the presence of photosynthetic traits, *F. tikoua* Bur. plants without CNPs exhibited the highest performance due to the highest subordinate function values. The function values decreased as the concentration of CNPs increased (Table 2). Regarding photosynthetic parameters (Table 3), *F. tikoua* Bur. plant showed improved tolerance to CNPs compared to the control group. Notably, for chlorophyll indexes, only plants exposed to 100 mg/kg CNPs had higher function values than the control group (Table 4).

## DISCUSSION

The exposure of *F. tikoua* Bur. plants to various concentrations of CNPs has been found to negatively impact their growth and photosynthetic processes. However, this inhibition is dose-dependent to a certain degree. As the concentration of CNPs increased, the plants exhibited decreased leaf area, branches numbers, and green leaf numbers compared to the control group, except for those treated with 50 g/kg CNPs, suggesting this inhibition may be attributed to high concentrations of CNPs. *Jackson et al. (2013)* demonstrated that, at

**Table 3 The function values of photosynthetic parameters in *F. tikoua* Bur.plant exposed to CNPs.**

| CNPs (mg/kg) | Function values of indicators | | | | | | | | Sum | Order |
|---|---|---|---|---|---|---|---|---|---|---|
| | E (molm$^{-2}$ s$^{-1}$) | A (μmolm$^{-2}$ s) | Ci (μmolmol$^{-1}$) | WUE (μmolmol$^{-1}$) | VPD (Kpa) | gtw (molm$^{-2}$ s$^{-1}$) | T (°C) | RH (%) | | |
| 0 | 1 | 1 | 1 | 0.79 | 0 | 0 | 0 | 0 | 0.47 | 4 |
| 50 | 0.33 | 0.73 | 0.85 | 1 | 1 | 1 | 1 | 0.5 | 0.8 | 1 |
| 100 | 0.1 | 0.04 | 0.53 | 0 | 0.97 | 0.89 | 0.99 | 0.99 | 0.56 | 2 |
| 200 | 0 | 0 | 0 | 0.54 | 0.91 | 0.85 | 0.99 | 1 | 0.54 | 3 |

**Table 4 The function values of Chlorophyll indexes in *F. tikoua* Bur.plant exposed to CNPs.**

| CNPs (mg/kg) | Function values of indicators | | | | Sum | Order |
|---|---|---|---|---|---|---|
| | Chl a | Chl b | Chl t | Chl a/b ratio | | |
| 0 | 0.65 | 0.64 | 0.67 | 0.67 | 0.66 | 2 |
| 50 | 0.29 | 0.3 | 0.26 | 0.25 | 0.28 | 3 |
| 100 | 1 | 1 | 1 | 1 | 1 | 1 |
| 200 | 0 | 0 | 0 | 0 | 0 | 4 |

low doses, carbon nanotubes have beneficial effects on seeds, roots, and plant transport, while high doses have been associated with negative effects. The growth inhibition induced by CNPs is likely due to the interference of CNPs with the plants' physiological functions. CNPs may disrupt the plants' nutrient and water uptake through the roots due to their unique properties. CNPs were also found to inhibit plant antioxidant enzyme activity and accumulate ROS beyond the plant's tolerance (*Vithanage et al., 2017*). The excess ROS then causes oxidative stress and damages plant tissues and organs, leading to cell damage and inhibit normal growth of plant.

Furthermore, CNPs exposure were found to significantly reduce the content of total chlorophyll and chlorophyll b in plant leaves, which are essential pigments for photosynthesis. Correspondingly, most photosynthetic parameters such as photosynthetic rate (A), transpiration rate (E), and intercellular $CO_2$ concentration (Ci) declined with rising CNP levels, especially for Ci. This suggests the CNPs interfere with the plants' normal growth and development processes and photosynthetic functions. The decreased chlorophyll content and impaired photosynthesis could be a direct result of CNPs toxicity to plant cells (*Begum et al., 2012*; *Mondal et al., 2011*). Previous studies have shown that limited doses of MWCNTs or SWCNTs can be taken up by the roots and leaves of various plant species, such as wheat (*Xia et al., 2012*), and rapeseed (*Larue et al., 2012*). The interaction between CNPs and plants can affect plant physiology by generating reactive oxygen species (ROS), which can disrupt chlorophyll synthesis (*Begum & Fugetsu, 2012*). ROS can harm cellular components like chloroplasts and interfere with photosynthetic electron transport, leading to a decrease in photosynthetic pigments and rates. This suggests that exposure to CNPs has the potential to impact crucial plant biological processes depending on the level of exposure.

The interaction between CNMs and plant cells is complex and multifaceted. The physiochemical properties of CNMs, such as size, shape, and surface charge, can influence their uptake, translocation, and potential effects on cellular processes. *Giraldo et al. (2014)* demonstrated the passive transport and localized accumulation of SWCNTs within the lipid envelope of chloroplasts extracted from spinach leaves. This integration enhanced the photosynthetic capabilities of the chloroplasts, introducing new functional properties. Interestingly, SWCNTs with high negative or positive zeta potentials (strongly cationic or anionic surface functionalization) were observed to localize more rapidly within the chloroplasts. The research by *Zhai et al. (2015)* investigated the effects of charge and size on the uptake and translocation of MWCNTs in maize and soybean plants. Three types of MWCNTs were examined: neutral, negatively charged (COOH-MWCNTs), and positively charged (NH2-MWCNTs). The study revealed cellular, charge, and size selectivity in the uptake and translocation of these MWCNTs in both plant species, suggesting that these properties may be crucial for the development of nanotransporters.

On account of subordinative function value, it showed that morphological characteristics, photosynthetic parameters, and chlorophyll content each had varying impacts on the tolerance of *F. tikoua* Bur. plants to CNPs. According to morphological parameters, plant tolerance to CNPs decreased with the increase of CNPs concentration, which can be explained by the fact that higher CNPs concentration leads to more accumulation of CNPs in plant tissues that interfere with plant growth (*Verma et al., 2019*). The strong positive correlation observed between BN, LA, and GL emphasizes the importance of these parameters in evaluating the inhibitory impact of CNPs on plant growth. This correlation highlights the interrelationship among these parameters and their potential contribution to elucidating the mechanisms underlying CNP-mediated inhibition. While physiological indexes like photosynthetic rates and chlorophyll levels influenced tolerance through different mechanisms related to metabolic processes. Specifically, higher photosynthetic rates and chlorophyll content help plants carry out photosynthesis more efficiently to overcome stress from CNP exposure (*Afzal & Singh, 2022*). These results align with previous work documenting the complex interplay between a plant's physical form and its functional biochemistry in determining stress responses (*Chugh, Kaur & Gupta, 2011*; *Kranjc & Drobne, 2019*). The research provides insight into how plant growth and physiology properties impact CNPs tolerance through different pathways. Understanding these mechanisms is important for assessing plant health and productivity under nanomaterial exposure (*Verma et al., 2018*; *Khalid et al., 2022*). The functionalization of carbon nanotubes (CNTs) with suitable molecules has been shown to mitigate their potential toxic effects, according to recent studies (*Pawar, Anumalla & Sharma, 2023*) . In addition, modifications to CNTs can enhance their ability to efficiently adhere to nucleic acids.MWCNTs have also been recognized as crucial in the delivery of DNA into the cytoplasm. Given the importance of F. tikoua Bur. for human health, it is crucial to investigate the potential toxic effects of carbon nanoparticles on the fruit of this plant. Future research should focus on the effects of various nanomaterials on the growth and development of F. tikoua Bur., in order to further elucidate the underlying mechanisms by which nanomaterials interact with this plant species.

## CONCLUSION

Our study findings indicated that the application of CNPs has a negative impact on the growth of *F. tikoua* Bur. plants by disrupting the photosynthetic physiological processes in a dose-dependent manner. It is also suggested that a concentration of 50 g/kg may be the threshold at which CNPs exhibit toxicity towards plants. Future research should assess plant responses at various CNPs concentrations to determine safe levels for application. Advanced analytical techniques should be employed to quantify CNP levels in various plant organs, such as leaves, stems, and fruits, and elucidate the mechanisms governing their accumulation and translocation. Additionally, toxicological studies are necessary to assess potential health risks associated with CNP exposure through Ficus tikoua Bur. consumption. This comprehensive approach will provide a thorough safety evaluation for human consumption. Understanding the mechanisms of CNPs phytotoxicity is crucial for the responsible use of nanomaterials in agricultural production.

### Funding

This work was supported by the program for Natural Science Research in Guizhou Education Department (QJJ-[2023]-024), the Sixth Batch of Guizhou Province High-level Innovative Talent Training Program (GCC [2022] 009), the program of Excellent Innovation Talents in Guizhou Province (GCC[2023]071), and National Natural Science Foundation of China (32160086, 31900271), Projects of Guizhou Provincial Science and Technology (QKHJC-[2019]1455). The funders had no role in study design, data collection and analysis, decision to publish, or preparation of the manuscript.

### Grant Disclosures

The following grant information was disclosed by the authors:
Natural Science Research in Guizhou Education Department: QJJ-[2023]-024.
The Sixth Batch of Guizhou Province High-level Innovative Talent Training Program: GCC [2022] 009.
The program of Excellent Innovation Talents in Guizhou Province: GCC[2023]071.
National Natural Science Foundation of China: 32160086, 31900271.
Projects of Guizhou Provincial Science and Technology: QKHJC-[2019]1455.

### Competing Interests

The authors declare there are no competing interests.

### Author Contributions

- Nian Chen performed the experiments, analyzed the data, prepared figures and/or tables, authored or reviewed drafts of the article, and approved the final draft.
- Xiaojian Tian performed the experiments, analyzed the data, prepared figures and/or tables, authored or reviewed drafts of the article, and approved the final draft.

- Mingli Yang performed the experiments, analyzed the data, prepared figures and/or tables, authored or reviewed drafts of the article, and approved the final draft.
- Jiajun Xu performed the experiments, analyzed the data, prepared figures and/or tables, and approved the final draft.
- Tinghong Tan conceived and designed the experiments, authored or reviewed drafts of the article, funding acquisition, and approved the final draft.
- Jiyue Wang conceived and designed the experiments, analyzed the data, authored or reviewed drafts of the article, and approved the final draft.

## Data Availability

The raw data are available in the Supplemental File.

## Supplemental Information

Supplemental information for this article can be found online at http://dx.doi.org/10.7717/peerj.17652#supplemental-information.

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
