# Peer review of "Effect of carbon nanoparticles on the growth and photosynthetic property of *Ficus tikoua* Bur. plant"

_PeerJ, doi:10.7717/peerj.17652_

## Round 0.1 · original submission · Major Revisions

· Academic Editor

Major Revisions

Please ensure that all the reviewers' comments are taken into consideration and that a revised version and a rebuttal letter are submitted.

Reviewer 1 ·

Basic reporting

Dear authors,
I reviewed the manuscript "Effect of carbon nanoparticles on the growth and photosynthetic property of Ficus tikoua Bur. plant". I found it well-written, but it has some flaws that can be easily corrected. The experimental design is appropriate, and the results are solid. However, it isn't ready for publication in its current form. Some issues must be corrected and justified in-depth, based on the following comments:

1) The background does not justify why CNP should be administered to the Ficus tikoua Bur plant. More than just wanting to know the effect of spherical CNPs on morphology and photosynthetic parameters is required.
2) There is no specific research question.
3) Is Ficus tikoua a model plant? If so, it should be clarified.
4) Furthermore, if carbon nanotubes were used in all the studies cited in the background, why were spherical nanoparticles used in this work?
5) Something of concern is that there is no explicit justification for administering carbon nanoparticles to plants in general. If nanoparticles translocate from roots to stems and leaves, could they also be translocated to fruits? In any case, the ultimate purpose of all these plants is human consumption. It is well known that some non-functionalized carbon nanoparticles can be very toxic to animal cells; why was this topic not discussed?

I suggest this reading:
https://doi.org/10.1016/B978-0-12-819786-8.00023-2

6) Another thing that stands out is that a current topic should have the newest references; however, in the manuscript, only 20% are under five years old.

Other comments are the following:
L37. The acronym UMCNT, is it a typo? It had not been defined previously.
L 40. When you say: "This suggests that....etc." It needs to be clarified why this suggestion is made if no results are presented from the algae growth test.
L52. Please correct "....Bur.palnt." I assume you meant "Bur. plant,"
L69. The writing has flaws; please correct it. Perhaps the conjunction AND before "...water use..." is unnecessary; if this is the case, a comma would be missing, and the word Leaf would be written in lower case.
L94. Why is it intriguing that there are no significant differences between CNP levels? There is no possible explanation?
L126. Table 3 does not show that the contents of chlorophylls a and b are reduced at a higher concentration of CNPs. Rather, the presence of CNPs, regardless of their concentration, significantly affected the chlorophyll a and b contents.
L132. The way this is written is confusing: ...CNTs or MWCNTs, SWCNTs...
L143. Please correct "he strong...etc." to "The strong...etc."

Experimental design

No comment

Validity of the findings

No comment

Additional comments

No more comments

·

Basic reporting

a. The language in the article is generally clear and simple, but some minor grammatical errors are present which must be checked. In general, the article text is technically correct.
b. Acronyms are not clearly defined and there are some errors and inconsistencies in acronym definitions (e.g., UWCNT, UMCNT, MWCNT, etc.). These errors must be addressed.
c. The introduction and discussion sections need include significantly more citations of CNTs and their applications as well as more citations to support the findings and conclusions.
d. In general, the tables are well represented in the raw data file. However, plotted data is missing and it is strongly recommended that the authors utilize figures to better visualize effects and trends as well as statistical analysis.

Experimental design

a. The article is within the scope of the journal.
b. The article does not clearly define the research questions. For example the authors state in the introduction: "The aim was to understand the interactions between nanomaterials and plants." This last statement in the introduction should be further expanded to very clearly detail all the research questions proposed and how the studies proposed help answer these questions.
c. The methods described in the article are not clear and lack major details of how assays were performed for repeatability. At the very least, the authors should point to literature detailing the methods utilized. I recommend that the methods sections be more detailed and broken up by assay to include test details for ease of replication of these studies.

Validity of the findings

a. In general, the data presented in the manuscript is robust and is a solid base for the conclusions. Statistical analyses are adequate, but more robust comparisons may be helpful including more discussion of statistical results. The relevant correlation plots should be generated for appropriate visualization of the correlations. In general, experimental controls are present.
b. The conclusions from the studies are somewhat well presented and discussed. The results do not necessarily describe the mechanism of the observed effects but the results are well described. There is some discussion to attempt to explain the observed phenomena, but it is difficult to justify it without a thorough investigation and further experiments. At minimum, the authors should outline further investigations to explain the results.

Additional comments

1. The introduction needs to include significantly more citations of CNTs and their applications.
2. What is the rational behind the choice for the FIcus Tikoua plant?
3. In order to arrive at a mechanistic understanding of the effects of CNTs on plant growwth metrics, ROS and other relevant metrics should be included. Can cell toxicity be assessed as well?
4. The effect on plant cells could also stem from the CNT physiochemical properties, e.g. size, shape, etc. Can the authors include discussion of those considerations from the literature and compare to the specific CNTs used in this study?

---

## Round 0.2 · accepted · Accept

· Academic Editor

Accept

Thanks for improving the manuscript.

Reviewer 1 ·

Basic reporting

I reviewed the revised version of the manuscript and believe that the suggested changes were made or a response was given accordingly.

Experimental design

No comments

Validity of the findings

No comments

Additional comments

No comments